# Multi-Strain Probiotic Supplementation with a Product Containing Human-Native *S. salivarius* K12 in Healthy Adults Increases Oral *S. salivarius*

**DOI:** 10.3390/nu13124392

**Published:** 2021-12-08

**Authors:** Karina Cernioglo, Karen M. Kalanetra, Anna Meier, Zachery T. Lewis, Mark A. Underwood, David A. Mills, Jennifer T. Smilowitz

**Affiliations:** 1Department of Food Science and Technology, University of California Davis, Davis, CA 95616, USA; kkurudimov@ucdavis.edu (K.C.); kmkalanetra@ucdavis.edu (K.M.K.); anmeier@ucdavis.edu (A.M.); damills@ucdavis.edu (D.A.M.); 2Foods for Health Institute, University of California Davis, Davis, CA 95616, USA; 3Synbiotic Health, Inc., Lincoln, NE 68508, USA; zactlewis@gmail.com; 4Department of Pediatrics, University of California Davis Children’s Hospital, Sacramento, CA 95817, USA; munderwood@ucdavis.edu; 5Department of Viticulture and Enology, University of California Davis, Davis, CA 95616, USA

**Keywords:** oral microbiome, probiotics, *S. salivarius* K12

## Abstract

*Streptococcus salivarius* (*S. salivarius*) K12 supplementation has been found to reduce the risk of recurrent upper respiratory tract infections. Yet, studies have not reported the effect of supplementation on oral *S. salivarius* K12 levels or the salivary microbiome. This clinical trial was designed to determine how supplementation with *S. salivarius* K12 influences the oral microbiome. In a randomized, double-blind, placebo-controlled trial, 13 healthy adults received a probiotic powder (PRO) containing *Lactobacillus acidophilus*, *Bifidobacterium lactis*, and *S. salivarius* K12 and 12 healthy adults received a placebo-control powder (CON) (*n* = 12) for 14 consecutive days. Oral *S. salivarius* K12 and total bacteria were quantified by qPCR and the overall oral microbiome was measured using 16S rRNA amplicon sequencing. Supplementation significantly increased mean salivary *S. salivarius* K12 levels by 5 logs compared to baseline for the PRO group (*p* < 0.0005), which returned to baseline 2 weeks post-supplementation. Compared with the CON group, salivary *S. salivarius* K12 was 5 logs higher in the PRO group at the end of the supplementation period (*p* < 0.001). Neither time nor supplementation influenced the overall oral microbiome. Supplementation with a probiotic cocktail containing *S. salivarius* K12 for two weeks significantly increased levels of salivary *S. salivarius* K12.

## 1. Introduction

Probiotics—defined by the International Scientific Association for Probiotics and Prebiotics as “live microorganisms that, when administered in adequate amounts, confer a health benefit on the host”—are widely used to support the health of the gastrointestinal tract [1]. Most probiotic strains are selected for, delivered to, and investigated for their impact on the gastrointestinal tract, although probiotic strains may be targeted to reside in and/or benefit other organ systems such as the reproductive tract, oral cavity, lungs, skin, and gut–brain axis. For example, *Streptococcus salivarius*, a prominent member of the oral microbiota, has been identified as a safe and effective probiotic targeting oral health [2].

Populations of *S. salivarius* represent 10^6^–10^7^ CFU/mL in saliva and 2% of the total streptococci isolated from the buccal mucosae, 17% from the tongue, and 30% from the pharynx [3]. A specific strain of *S. salivarius* (*S. salivarius*), K12, is commercially available for targeting the health of oral mucosal surfaces. A small number of studies have shown associations between oral supplementation with *S. salivarius* K12 and the health of the ear and oral cavity. The putative health functions of *S. salivarius* K12 are proposed to occur via several mechanisms: (1) displacing pathogens; (2) producing lantibiotics (a subgroup of bacteriocins), antibiotic peptides that target Gram-positive microbes including pathogens, and (3) modulating immune responses of human mucosal cells. For example, bacteriocin-producing *S. salivarius* K12 was found to temporarily displace salivary anaerobes that produce valeric and butyric acids or putrescines which are responsible for oral malodor [4]. Additionally, *S. salivarius* K12 is well known to produce the lantibiotics salivaricin A (SalA) and salivaricin B (SalB), that inhibit the growth of *S. pyogenes* [5,6]. Moreover, adherence of *S. salivarius* K12 to human nasopharyngeal epithelial cells was found to upregulate anti-inflammatory responses by these cells, thereby potentially offering host protection from inflammation and apoptosis induced by pathogens [7]. The proposed mechanisms of action by *S. salivarius* K12 on oral health have been based on in vitro studies and have not been confirmed in in vivo studies.

A recent systematic review on probiotic supplementation with *S. salivarius* K12, summarizing nine studies, reported that *S. salivarius* K12 supplementation reduced the occurrence and/or severity of secretory otitis media in children and also prevented streptococcal and pharyngotonsillitis in adults and children [8]. All of the studies described in the systematic analysis were observational or open, non-randomized trials. To date, no studies on the impact of *S. salivarius* K12 supplementation on any health outcomes or effects of the oral microbiome have been double-blind, placebo-control trials. While in vitro and observational data suggest that the K12 strain may be a good candidate for improving human ear and oral cavity health, double-blind, randomized, placebo-controlled clinical trials are warranted before any specific clinical recommendations can be proposed. The purpose of the UC Davis Probiotic Oral Health (PROHealth) Study was to determine the effect of *S. salivarius* K12 supplementation on salivary *S. salivarius* K12 levels and the salivary microbiome compared with a placebo-control in healthy adult participants.

## 2. Materials and Methods

### 2.1. Study Population

Between September 2018 and January 2019, twenty-six healthy adults who lived within Yolo and Sacramento Counties in California provided written informed consent to participate in this study. Enrollment criteria aimed to limit confounding variables that could potentially influence the oral microbiome. Inclusion and exclusion criteria were as follows: adults aged 21–45 years, who practiced good oral hygiene according to American Dental Association Guidelines by brushing teeth between 1–2 times per day, with no history of periodontal disease or gingivitis, no history of oral surgery, intensive procedures, dental trauma or injury, or routine dental cleanings within the past 4 weeks, no history of tobacco use within the past year, no history of marijuana or illicit drug use within the past 4 weeks, no history of antibiotic use or use of probiotics containing *S. salivarius* within the past 8 weeks, no history of consumption of probiotics, kombucha, yogurt, or kefir within the past week, and who had not been diagnosed with any chronic metabolic disease or obesity.

### 2.2. Study Design

The PRO Health Study was a double-blind, placebo-control, prospective five-week trial. The University of California Davis Institutional Review Board approved all aspects of the study (IRB #: 1188050-8). This trial was registered on ClinicalTrials.gov Identifier: NCT03748017.

Prior to initiation of the study, the probiotic powder was randomized by the study sponsor via block randomization in six blocks of four participants per block. Each block was allocated to either all males or all females to control for the effect of gender on oral hygiene practices and to achieve an equal gender ratio between active and placebo arms. A commercial formula containing both an *S. salivarius* native to the oral cavity and two gastrointestinal-targeted strains was provided for use by the study sponsor. Each probiotic powder sachet (1.5 g) contained approximately 7.77 billion CFU of *L. acidophilus* DDS-1^®^, 8.25 billion CFU of *B. lactis* UABla-12, and 2 billion CFU of *S. salivarius* BLIS K12^TM^, xylitol (943 mg), inulin (450 mg), fruit punch and cherry flavors (approximately 16.5 mg), and inert manufacturing aids (silica, approximately 15 mg). The placebo contained additional excipient in place of the probiotic content, but was otherwise identical. Participants were randomized into the active or placebo arms following a final screening. Supplementation occurred on day 8 through day 21, during which participants consumed one daily dose (one 1.5 g sachet) of probiotic powder or placebo every evening, directly by mouth without mixing with any liquid prior to consumption. Participants were instructed to consume the supplement at least 30 min after brushing their teeth and to refrain from consuming food or beverages and using oral care products until the following morning. However, intake of water was permitted after at least 30 min had passed following consumption of the supplement. Participants were instructed to keep all sachets in a cool dry place until consumption and to return all used and unused sachets to study personnel, which were counted to assess compliance. Non-compliance was defined as <75% of supplementation (fewer than 11 sachets consumed).

To reduce confounding variables, participants were asked to use a provided toothpaste (Tom’s of Maine Wintermint Whole Care Toothpaste, Lot Number: 8142UST11C) for the duration of the study period. Throughout the five-week study, participants were instructed to avoid consuming probiotics, kombucha, yogurt, and more than 1 package (approximately 15 pieces) of sugary candy per day, such as hard or gummy candy or mints; to avoid chewing more than 1 package of chewing gum (approximately 15 pieces) per day; to avoid using non-study oral-care products, except for flossing, such as non-study provided toothpaste, mouthwash, and breath-sprays; and to avoid consuming alcohol 24 h prior to each sample. Participants were also asked to limit consumption of raw onions or garlic and fermented foods during the study period.

The five-week study period consisted of a one-week lead-in period, during which baseline salivary samples were collected, a two-week intervention phase, during which participants were supplemented with either the probiotic powder or placebo and salivary samples were collected, and a two-week observation phase, during which salivary samples continued to be collected. During the first week of the study, participants completed a health history questionnaire to gather information about their demographics, general health, diet, lifestyle, and oral health history. On study day 7 and 37, participants were weighed using a digital bathroom scale (Tanita, Shanghai, China). Throughout the duration of the study, participants were required to complete a Daily Health Log to report illness, antibiotic use, intake of medications, and stooling outcomes, including the number of stools per day and the firmness and consistency of each first stool of the day, based on a scale of 1–10 where 1 is “extremely watery, almost entirely liquid” and 10 is “extremely hard, difficult to pass”, and the Bristol Stool Scale, respectively; a Daily Intake Log to report the consumption of confounding variables of the microbiome; and a Daily Oral Health Log to report their daily oral hygiene routine and non-study oral care product use.

### 2.3. Power Analysis

Using preliminary data from a pilot study (IRB #: 1188050-8), this study required randomization of twenty participants to identify a 7-log difference in salivary *S. salivarius* with α = 0.01 (to account for multiple testing within each family of hypotheses), and power = 90%. The variance was calculated as half the standard deviation from saliva samples collected on the third day of supplementation. Twenty-six individuals were enrolled to account for attrition.

### 2.4. Sample Collection

Salivary swab samples were collected using a Puritan swab (Puritan Medical Products Inc., Hayward, CA, USA) and a Zymo tube containing stabilization buffer (Zymo Research Corporation, Irvine, CA, USA). Participants were instructed to collect samples upon waking up and prior to eating or drinking any beverages, including water, brushing teeth, or using oral care products. Oral biofilm was collected by rubbing the swab between the cheek and lower gum for 30 s and the back of the tongue for 15 s while rotating the shaft. The swabs were placed into the Zymo tubes which were subsequently inverted six times. Collected samples were stored in a cool, dry place until a scheduled visit with study coordinators on days 7, 22, and 37. Then the salivary swab samples, without labeled group assignments, were stored at −80 °C until processed for DNA extraction. Group assignments were unblinded, post-microbial analysis.

### 2.5. DNA Extraction

Genomic DNA from a total of 73 oral swabs was extracted using Zymobiomics DNA Miniprep Kit (Zymo Research Corporation, Irvine, CA, USA) per the manufacturer’s instructions for swab samples. The Zymo stabilization buffer is also a lysis buffer so that swab samples did not need to be transferred to a new lysis buffer. Briefly, swabs were vortexed for 30 s to transfer microbes from the swab to the buffer and 1 mL of the buffer was transferred to ZR BashingBead lysis tubes (Zymo Research Corporation, Irvine, CA, USA). Samples were processed for a total of 5 min at 1 min intervals of 6.5 m/s in a FastPrep-24 Classic Beadbeater (MP Biomedicals, Irvine, CA, USA), cooling on ice for 5 min between homogenizations. The rest of the extraction was continued as stated in the Zymobiomics protocol.

### 2.6. QPCR

SYBR green and TaqMan quantitative polymerase chain reaction (qPCR) assays were performed on a 7500 Fast Real-Time PCR System (Applied Biosystems, Carlsbad, CA) with primers specific for universal bacteria [9] and *Streptococcus salivarius* K12 TaqMan primers and probe [3]. Universal bacterial SYBR green assays contained 10 μL of 2× SYBR Advantage QPCR Premix master mix (Clontech Laboratories, Inc., Mountain View, CA, USA), 5.6 μL water, 1 μL each forward and reverse primers (4 μM), 0.4 uL 50× ROX reference dye, and 2 μL genomic DNA. SYBR green cycling conditions were denaturation at 95 °C for 20 s followed by 40 cycles of 95 °C for 4 s and annealing at 65.5 °C for 25 s and a final melt curve. TaqMan assays contained 10 μL 2× TaqMan Universal PCR master mix (Applied Biosystems), 2 μL each forward and reverse primers (1 μM) and 1 uL TaqMan probe (1 μM), 3.5 μL water, and 1.5 μL genomic DNA. Reaction conditions for the TaqMan assay were denaturation at 95 °C for 20 s followed by 40 cycles of 95 °C for 1 s and annealing at 60 °C for 20 s. All reactions were carried out in triplicate with a nontemplate control (nuclease-free water) and a standard curve made of serial dilutions 1.69 × 10^1^ to 1.69 × 10^7^
*S. salivarius* K12 uL^−1^.

### 2.7. 16S rRNA Gene Amplicon Sequencing Library Preparation

The V4 region of the 16S rRNA gene was amplified in triplicate by targeted barcoded primers F515 (5′-NNNNNNNNGTGTGCCAGCMGCCGCGGTAA-3′) and R806 (5′-GGACTACHVGGGTWTCTAAT-3′) as previously described [10]. Amplicons were verified by gel electrophoresis, pooled, then purified with the Qiagen QIAquick PCR Purification Kit (Qiagen, Germantown, MD, USA) and taken to the UC Davis Genome Center DNA Technologies Sequencing Core for library preparation and 250 bp paired-end sequencing on an Illumina Miseq (Illumina, San Diego, CA, USA).

### 2.8. 16S rRNA Gene Amplicon Sequencing Data Analysis

Resulting raw data were demultiplexed with sabre [11] and then imported into QIIME2 (version QIIME2-2019.7) [12,13]. Bases before base pair 21 and after base pair 242 for the forward read, before base pair 20 and after base pair 250 for reverse read, were trimmed. Trimmed reads were processed with DADA2 [14]. Trimmed and filtered sequences were aligned, and taxonomy was assigned using the 99% SILVA naïve Bayes classifier [15] in QIIME2-2019.7. A midpoint rooted phylogenetic tree was inferred using Fasttree [16]. Weighted and unweighted UniFrac distance matrices were calculated based on the mid-point rooted phylogenetic tree and the DADA2 replicon sequence variant (RSV) table rarefied to a depth of 6000 reads. Distance matrix and RSV tables were exported from QIIME2 for further analysis in R and R Studio (version 3.6.1 and 1.2.335, respectively) and GraphPad Prism 8 version 8.2.1.

### 2.9. Statistics

Kruskal–Wallis was used to determine the effect of the intervention at each time point on salivary mean copy# of the *salA* gene/µg DNA (for *S. salivarius*) and salivary mean copy# of 16S rRNA gene/µg DNA (for universal bacteria). The Friedman test was used to determine the effect of time for each intervention on salivary mean copy# of the *salA* gene/µg DNA and salivary mean copy# of 16S rRNA gene/µg DNA. If time was significantly different according to the Friedman test, a paired Wilcoxon signed-rank test was used to compare median differences between each time point. Shannon entropy was calculated for alpha diversity and differences between PRO and CON groups were determined using Kruskal–Wallis pairwise testing at each time point. Permutational multivariate analysis of variance (PERMANOVA) [17], based on weighted and unweighted UniFrac distances, as implemented in the R VEGAN package, was used to determine if there were significant differences in microbial community structure between treatment groups. MaAslin2 was used to determine microbial features associated with the intervention and time using general linear models [18]. Time point and treatment were run as fixed effects and subject was run as a random effect for the model. Additionally, analysis of composition of microbiomes (ANCOM) was used to investigate differential abundance of taxa at the phylum, family, and genus levels [19]. Data for the primary outcomes measured by QPCR include all participants (intent-to-treat analysis) and 16S rRNA gene amplicon sequencing data include all participants who completed the protocol.

Daily stool frequency and firmness and the proportions of stool consistency, illnesses, and antibiotic use from the Daily Logs were averaged within each of the three time periods: baseline (Days 1–7), intervention (Days 8–21), and post-intervention (Days 22–36). Mann–Whitney U was performed to determine mean rank differences in stool frequency, stool firmness, and reported illness between groups at each time point. Mann–Whitney U was performed to determine a difference in the mean ranks of body weight between groups at baseline and at post-intervention. Descriptive statistics were performed on all collected demographic and health-related data.

## 3. Results

### 3.1. Study Participation

Thirty-eight adults were screened for eligibility to participate in the study. Twenty-six adults consented and were enrolled in the study, of which twenty-five met all final study criteria and were randomly assigned to active (*n* = 13) or placebo (*n* = 12) arms (Figure 1). Of the randomized participants, one participant from the active arm discontinued supplementation and consumed their last dose on Day 6 of the intervention period due to the development of hives on the abdomen, elbows, and knees. This participant remained in the study and their data are reported herein. A second participant from the active arm did not collect samples and withdrew from the study due to scheduling conflicts; therefore, they were not included in the final analyses. All other participants were compliant. The overall attrition rate for this study was 4%.

### 3.2. Salivary Microbes Measured by QPCR

Compared with the CON group, median salivary *S. salivarius* (mean copy# of *salA* gene/µg DNA) was significantly higher by 4.3 logs (*p* < 0.01, intent-to treat, Figure 2A) and 5 logs (*p* < 0.001, per-protocol, Figure 2B) in the PRO group during the intervention time point. The Friedman test identified significant differences in the distributions of *S. salivarius* over time in the PRO (*p* < 0.001) but not the CON group. In the PRO group, median salivary *S. salivarius* increased by 5 logs from baseline to the intervention time point (*p* < 0.01) and returned back to baseline levels during the post-intervention time point (intent-to-treat and per-protocol). Dot plots for salivary *S. salivarius* for each participant over the study period are shown in Appendix A. Salivary mean copy# of 16S rRNA gene/µg DNA (universal bacteria) was not different between intervention groups and did not change across time (Figure 3).

### 3.3. Salivary Microbiome Measured by 16S rRNA Amplicon Sequencing

Shannon entropy used to represent alpha diversity of the microbial communities was not different between the groups during the baseline, intervention, or post-intervention periods (Appendix A). According to PERMANOVA testing of both weighted and unweighted UniFrac distances, there were no significant differences of the salivary microbiome between groups or across time points (Figure 4). Additionally, neither MaAsLin2 nor ANCOM identified differences in any taxa by intervention group or time point, including the relative abundances of *Streptococcus*, *Bifibobacterium*, or *Lactobacillus*—the three genera in the probiotic product administered. Bar graphs for the salivary microbiome for each participant over the study period are shown in Appendix A.

### 3.4. Participant Health and Gastrointestinal Symptoms

There were no differences in participant weight between intervention groups or across time (Appendix A). Additionally, there were no differences in daily stool frequency, firmness, or consistency between groups (Table 1). None of the participants reported intake of antibiotics during the five-week study period. Participants in both groups reported the same number of days they experienced illness and the same number of participants reported feeling ill during the baseline and intervention period. Participants in the CON compared with the PRO group reported a higher percentage of illness days during the post-intervention period. Yet these differences were not significantly different and the number of illness days was low for both groups (Table 2). There were no differences in oral hygiene, use of the study toothpaste, non-study oral care products, or oral inhalers between the two groups (Table 3). None of the participants reported using non-study probiotics throughout the study period. There were no differences between the groups for the intake of fermented dairy or kombucha; garlic or onions; and sugary gum or candy (Table 4).

## 4. Discussion

This study is the first double-blind, randomized placebo-control trial to demonstrate that probiotic supplementation of *S. salivarius* K12 significantly increases levels of salivary *S. salivarius* K12. Additionally, probiotic supplementation did not alter salivary total universal bacteria. Supplementation with *S. salivarius* K12 for 2 weeks did not alter the salivary microbiome using 16S rRNA amplicon sequencing methodologies, including the relative abundance of *Streptococcus*. *S. salivarius* K12 supplementation for two weeks was well tolerated by the study population with the exception of one participant who experienced hives during supplementation which disappeared when supplementation ceased. 

Supplementation with *S. salivarius* K12 for two weeks significantly increased levels of salivary *S. salivarius* K12 by 5 logs during supplementation; however, mean levels were not significantly different two weeks post-supplementation, suggesting, on the whole, transient colonization. There was, however, large inter-individual variability in the detection of *S. salivarius* presence in our cohort. Similar to our findings, a recent randomized, open-label study found that probiotic supplementation with *S. salivarius* K12 increased salivary *S. salivarius* during the supplementation period but, on average, subjects returned to baseline levels when supplementation ceased [20]. The observation that a probiotic strain’s levels decrease dramatically or even disappear from the target site upon cessation of the probiotic (without the aid of a combination with a targeted prebiotic (e.g., a synbiotic), is consistent with the literature. Persistent high-level colonization with probiotic supplementation in adults has been inconsistent and highly variable [21]. Further study is needed to explain this “responder/non-responder” phenomenon with *S. salivarius* and other probiotics. 

Supplementation with *S. salivarius* K12 did not alter the total bacterial load in the mouth. This observation (taken in isolation) demonstrates that the increase of *S. salivarius* observed during the intervention period did not significantly increase total bacteria-colony-forming units but did allow for some compositional changes to the salivary microbiome. However, the 16S amplicon sequencing found that supplementation with *S. salivarius* K12 did not significantly alter the overall oral microbiome structure at relatively high taxonomic levels. Several other probiotic supplementation studies have shown that the overall microbiome may not change even though the levels of specific microbes found in the probiotic formulation increase in target body environments [22,23].

Potential confounders in the study include the multiple strain formulation and the low dose of the prebiotic in the formula (inulin). It is not anticipated that the residence time of the powder in the mouth would allow significant metabolism of the inulin by either the probiotic strains or other resident microbes in the mouth; however, inulin-free controls to test its impacts were not available. The possibility of a synbiotic effect between the strains and the prebiotic was not investigated.

Changes in the microbial community that suggest colonization by the other two (gastrointestinal-targeted) species in the probiotic formula were not observed. This was not unexpected as the other two strains in the product were not native to the oral cavity or adapted to that environment, and previous observations suggest probiotic strains are at a competitive disadvantage when administered to environments to which they are not native [24]. As changes to the oral community within the *Bifidobacterium* or *Lactobacillus* genera were not originally hypothesized, targeted methods such as qPCR were not employed to test that hypothesis. However, digestive symptoms were tracked in this study to investigate the potential gastrointestinal effects of the other two (gastrointestinal-targeted) strains in the study subjects, and no statistically significant changes were observed.

The possibility remains, however, that there are finer grain taxonomic changes taking place in the microbiome that are below the level of taxonomic discrimination of the sequencing methods used here. 16S rRNA gene amplicon sequencing does not have the resolution to identify some subtle species-specific (but potentially significant) changes within the microbiome during supplementation, and the method has limited predictive value for functional differences. As *S. salivarius* K12 supplementation increased levels of *S. salivarius* K12 by 5 logs, and total universal bacteria and the relative abundance of *Streptococcus* did not change, we would speculate that other *Streptococcus* species or strains are decreased but that these changes are not captured by 16S rRNA gene amplicon sequencing. Indeed, this hypothesis is congruent with a bacteriocin-dependent mechanism of action of the strain and the known narrow specificity of the salivaricin A and B lantibiotics to taxa closely related to *S. salivarius* in phylogeny [25].

A more suitable method for identifying changes within *Streptococcus* would include strain-level metagenomic profiling which was not conducted in this study. Another limitation to this study is a lack of functional analyses to confirm that *S. salivarius* K12 is active in the oral mucosa. For example, peptidomics analyses could be useful to measure the lantibiotics salivaricin A and B, which are known to be produced by *S. salivarius* K12; this may be of particular importance given the antibacterial, antiviral, anti-inflammatory, and immunomodulatory capacity of some bacteriocins [26].

One of our study’s strengths is the use of a double-blind randomized control design. All previous studies using *S. salivarius* K12 were open-label, non-randomized, or were without placebo-control arms. Another strength is that many of the human studies that reported the effects of *S. salivarius* on health or other outcomes used a lozenge to deliver the probiotic. Use of a lozenge has its limitations with respect to applications, and is costly due to the large number of starting bacteria needed to survive the heat and pressure during lozenge formulation and compression. The potential use of a powder for multiple applications to improve oral and immunological health in children and adults warrants further investigation of this probiotic in a well-designed human trial. While the downside of accidental aspiration of powder containing live bacteria exists in treatments delivered in this form, we did not receive any such complaints or accounts of related adverse events from subjects in this trial.

This study is the first double-blind, randomized, placebo-control trial to demonstrate significant increases in levels of salivary *S. salivarius* K12, a potentially effective clinical strategy for reducing otitis media and pharyngotonsillitis. Future studies are warranted to examine how probiotic supplementation with *S. salivarius* K12 influences microbial function and consequent health outcomes in vivo.

## Figures and Tables

**Figure 1 nutrients-13-04392-f001:**
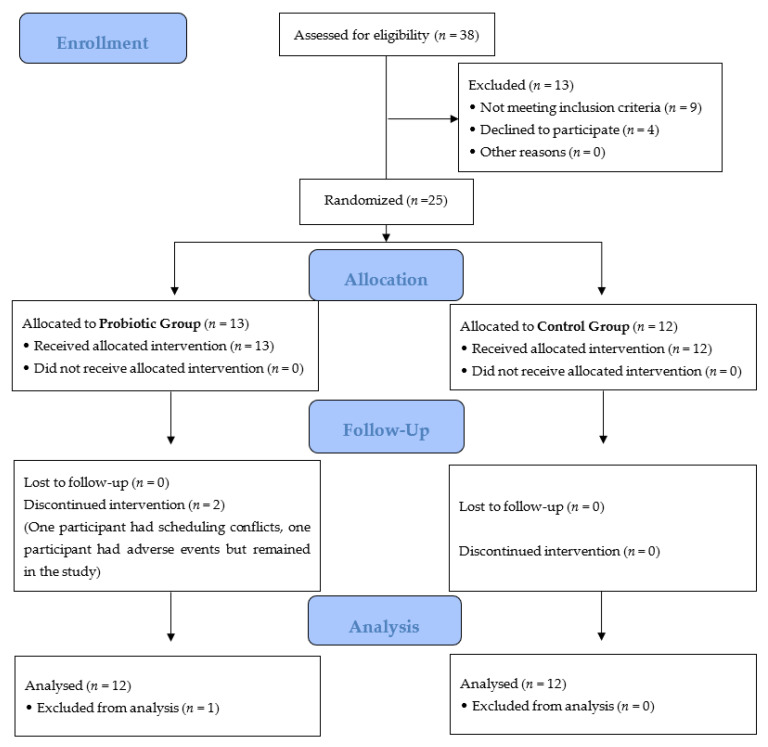
Consort diagram. Two participants discontinued the PRO intervention. One participant was a screen failure due to the use of antibiotics before the Day 7 randomization visit. One participant withdrew due to scheduling conflicts and the other participant experienced a rash in response to the supplement and consumed her last dose of the supplement on Day 6 of the intervention period but remained in the study. All other participants tolerated the supplement.

**Figure 2 nutrients-13-04392-f002:**
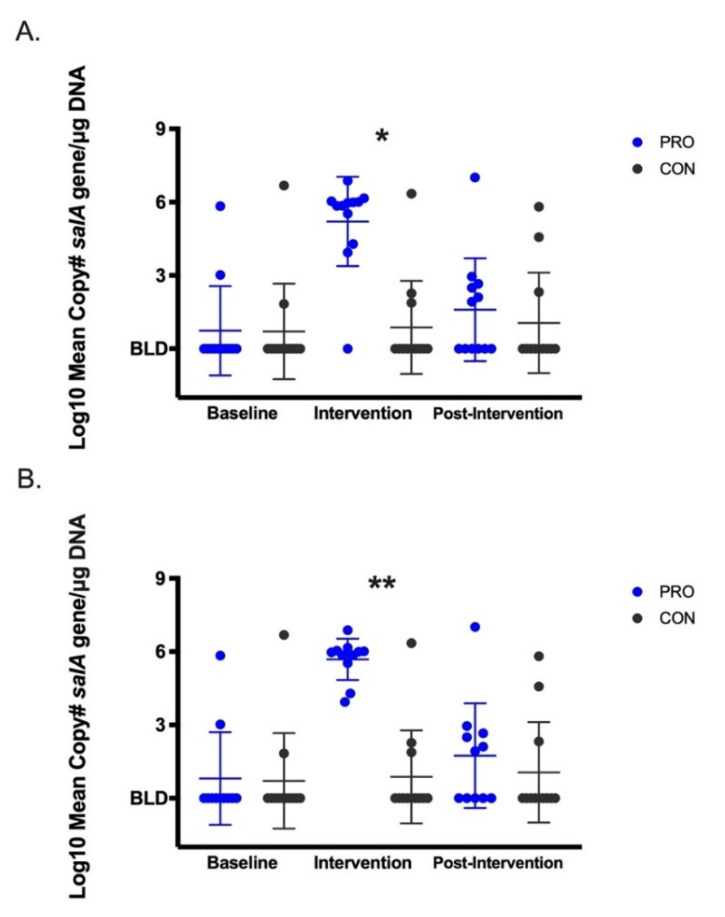
Log_10_ mean copy# *salA* gene (*S. salivarius*) per µg DNA. (**A**) Intent-to-treat with inclusion of participant 5003. (**B**) Per-protocol with participant 5003 removed from the analysis. * *p* < 0.01, ** *p* < 0.001. Dot plot represents each data point with mean ± SD. Below detection limit, BLD.

**Figure 3 nutrients-13-04392-f003:**
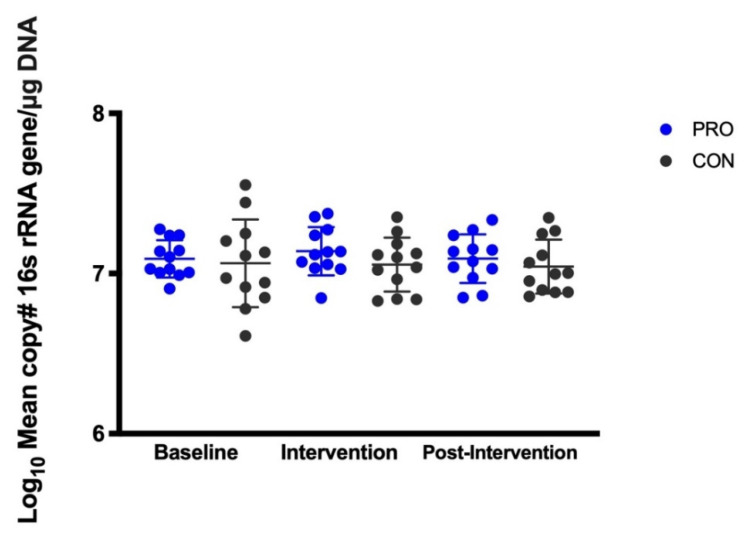
Log_10_ mean copy # 16S rRNA gene (total bacteria)/per µg DNA. Intent-to-treat with inclusion of participant 5003. Dot plot represents each data point with mean ± SD.

**Figure 4 nutrients-13-04392-f004:**
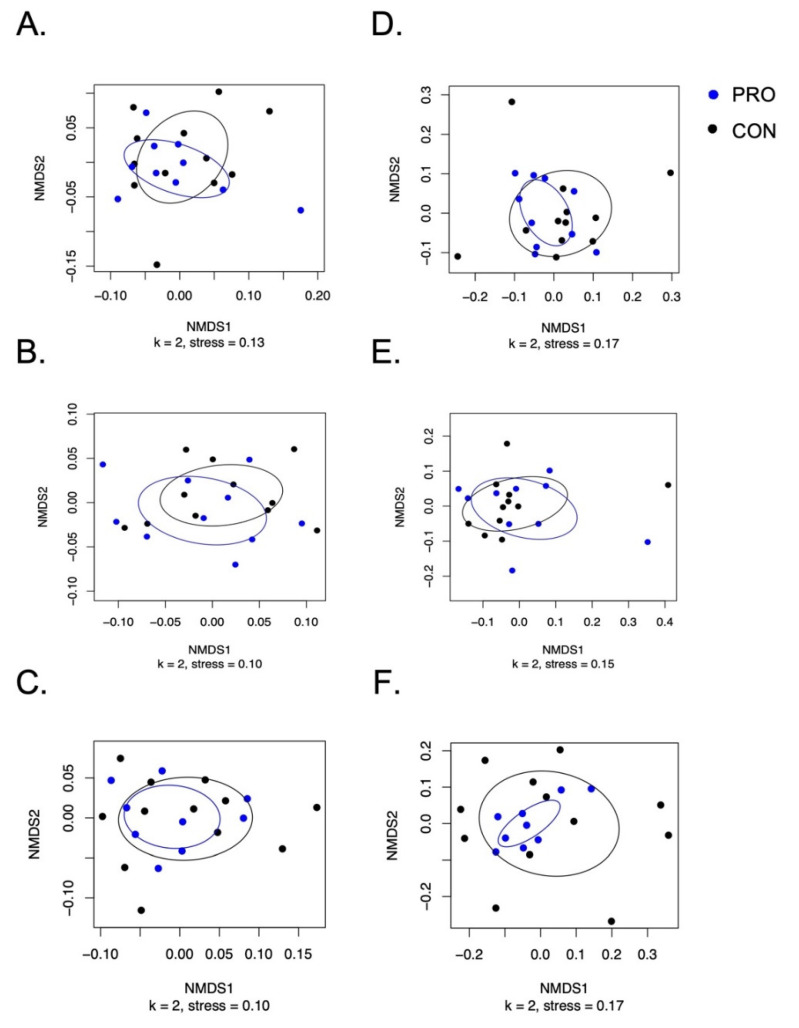
Non-metric multi-dimensional scaling plots for the salivary microbiome. Weighted and unweighted UniFrac NMDS plots with points colored by treatment (blue PRO were treated with probiotic and black CON were control). Weighted UniFrac NMDS plots: (**A**) baseline; (**B**) intervention; (**C**) post-intervention. Unweighted UniFrac NMDS plots: (**D**) baseline; (**E**) intervention; (**F**) post-intervention. Ellipses were drawn based on the standard deviation of the points within the respective intervention group. PERMANOVA testing between treatment groups did not reveal significant differences between the microbiota. Data shown are per-protocol with participant 5003 removed from the analysis.

**Table 1 nutrients-13-04392-t001:** Stool characteristics.

	PRO						CON					
	(*n* = 12)						(*n* = 12)					
	Baseline		Intervention		Post-Intervention		Baseline		Intervention		Post-Intervention	
	Mean	SD	Mean	SD	Mean	SD	Mean	SD	Mean	SD	Mean	SD
Number of Stools (# per day)	1.5	0.55	1.3	0.51	1.3	0.56	1.8	0.82	1.8	0.85	1.7	0.78
Stool Firmness (Rating 1–10)	5.2	0.94	4.9	0.52	4.9	0.59	5.0	1.1	4.8	0.84	4.7	0.92
Stool Consistency (Rating Type 1–7), (% Rating):												
Type 1	9.5%	23.1%	6.0%	11.3%	7.2%	25.0%	4.8%	12.7%	1.2%	2.8%	1.7%	3.0%
Type 2	6.0%	11.3%	4.8%	8.8%	2.2%	7.7%	4.8%	9.3%	12.5%	26.7%	3.3%	5.3%
Type 3	17.9%	16.3%	15.5%	16.1%	10.0%	13.5%	29.8%	27.6%	20.8%	21.8%	16.1%	14.6%
Type 4	33.3%	23.1%	32.1%	19.4%	53.3%	29.3%	33.3%	30.0%	45.2%	27.1%	58.9%	23.4%
Type 5	9.5%	14.1%	19.6%	17.3%	11.7%	17.1%	15.5%	21.5%	11.3%	15.7%	13.3%	18.2%
Type 6	13.1%	22.3%	8.9%	16.7%	4.4%	8.2%	4.8%	11.1%	4.8%	10.3%	2.8%	7.8%
Type 7	0.0%	0.0%	0.0%	0.0%	0.0%	0.0%	0.0%	0.0%	0.6%	2.1%	0.0%	0.0%
NA (No stool to rate)	10.7%	16.3%	11.3%	16.5%	11.1%	21.3%	7.1%	12.9%	3.6%	4.8%	3.3%	8.3%

# represents number.

**Table 2 nutrients-13-04392-t002:** Participant health.

	PRO						CON					
	(*n* = 12)						(*n* = 12)					
	Baseline		Intervention		Post-Intervention		Baseline		Intervention		Post-Intervention	
	Mean	SD	Mean	SD	Mean	SD	Mean	SD	Mean	SD	Mean	SD
Antibiotic Use, (% of # days)	0.0%	0.0%	0.0%	0.0%	0.0%	0.0%	0.0%	0.0%	0.0%	0.0%	0.0%	0.0%
Illness, (% # days)	2.4%	5.6%	2.4%	6.3%	3.3%	8.3%	3.6%	8.9%	2.4%	6.3%	12.8%	22.3%
Illness, (# of participants)	2		2		2		2		2		4	

# represents number.

**Table 3 nutrients-13-04392-t003:** Participant oral health.

	PRO (*n* = 12)						CON (*n* = 12)					
	Baseline		Intervention		Post-Intervention		Baseline		Intervention		Post-Intervention	
	Mean	SD	Mean	SD	Mean	SD	Mean	SD	Mean	SD	Mean	SD
Brush Teeth (# times/day)	1.9	0.4	1.9	0.4	1.9	0.4	1.9	0.3	2.0	0.4	1.9	0.4
Study Toothpaste Used, (% # days)	98.8%	4.1%	99.4%	2.1%	92.8%	21.0%	98.8%	4.1%	100.0%	0.0%	100.0%	0.0%
Non-Study Oral Care Products Use, (% # of days)	0.0%	0.0%	0.0%	0.0%	6.7%	21.1%	1.2%	4.1%	0.0%	0.0%	0.0%	0.0%
Oral Inhaler Use, (% # of days)	8.3%	28.9%	5.4%	18.6%	5.0%	17.3%	1.2%	4.1%	0.0%	0.0%	0.0%	0.0%

# represents number.

**Table 4 nutrients-13-04392-t004:** Participant intake of potential confounders of the salivary microbiome.

	PRO (*n* = 12)						CON (*n* = 12)					
	Baseline		Intervention		Post-Intervention		Baseline		Intervention		Post-Intervention	
	Mean	SD	Mean	SD	Mean	SD	Mean	SD	Mean	SD	Mean	SD
Non-Study Probiotics, (% # of days)	0.0%	0.0%	0.0%	0.0%	0.0%	0.0%	0.0%	0.0%	0.0%	0.0%	0.0%	0.0%
Yogurt, Kombucha, or Kefir Use, (% # of days)	0.0%	0.0%	0.6%	2.1%	1.7%	4.1%	0.0%	0.0%	0.0%	0.0%	0.0%	0.0%
Raw Onion or Garlic, (% # of days)	4.8%	9.3%	7.1%	15.8%	6.7%	9.0%	6.0%	7.4%	2.4%	6.3%	6.1%	10.4%
Gum, (% # of days)	16.7%	33.8%	15.5%	26.2%	15.6%	24.3%	4.8%	12.7%	3.6%	10.3%	2.8%	7.8%
Sugary Candy, (% # of days)	10.7%	13.8%	14.9%	12.4%	12.2%	14.2%	15.5%	24.7%	18.5%	20.3%	8.3%	15.3%

# represents number.

## Data Availability

Not applicable.

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
