# Peer review of "Multi-Strain Probiotic Supplementation with a Product Containing Human-Native S. salivarius K12 in Healthy Adults Increases Oral S. salivarius"

_nutrients, 2021, doi:10.3390/nu13124392_

Round 1

Reviewer 1 Report

The paper is well done and results clearly show the capability of the strain to colonize oral microbiota.

Nevertheless, I suggest the Authors to clearly explain the fact of using a mixed formula containing other 2 strains without showing the results of their possible colonization. Negative or positive, this important since S. salivarius is a typical oral taxon and make sense it could colonize. But what about probiotics do not really belong to the oral microbiota? In case of positive colonization of them it could be that any strain could colonize temporarly the oral mucosa and saliva? In case of negative results, this means that being a oral strain give a clear advantage. So, I kindly ask the Authors to explain in the paper why they did not provide this info concerning the other 2 strains.

I guess that another important comment to write on the paper would the fact of using a sequence homology of 99%. Could it be that using Amplicon sequence variants insted of OTU should have allowed to see a modification of the oral consortium? please add a specific comment on that.

Author Response

Reviewer 1

The paper is well done and results clearly show the capability of the strain to colonize oral microbiota.

Nevertheless, I suggest the Authors to clearly explain the fact of using a mixed formula containing other 2 strains without showing the results of their possible colonization. Negative or positive, this important since S. salivarius is a typical oral taxon and make sense it could colonize. But what about probiotics do not really belong to the oral microbiota? In case of positive colonization of them it could be that any strain could colonize temporarly the oral mucosa and saliva? In case of negative results, this means that being a oral strain give a clear advantage. So, I kindly ask the Authors to explain in the paper why they did not provide this info concerning the other 2 strains.

We thank the reviewer for the astute comments. Our sequencing data did not show changes in the microbial community that suggest a significant impact by the other two species in the probiotic formula. This negative result was not unexpected as the other two strains in the product were not native to the oral cavity or adapted to that environment, and previous observations suggest non-native probiotic strains are at a competitive disadvantage (http://dx.doi.org/10.4236/aim.2012.23051). The probiotic formula supplied to us by the study sponsor contained the other two strains for commercial reasons. As we did not originally hypothesize any changes within the Bifidobacterium or Lactobacillus genera, targeted methods such as qPCR were not employed to further test that hypothesis. Language has been added to the manuscript within the discussion and results sections to clarify these points and highlight the native origin of the S. salivarius strain.

I guess that another important comment to write on the paper would the fact of using a sequence homology of 99%. Could it be that using Amplicon sequence variants insted of OTU should have allowed to see a modification of the oral consortium? please add a specific comment on that.

We thank the reviewer for highlighting any potential confusion about our method. We did not report OTUs but rather RSVs. The microbiome community has been moving more and more towards using RSVs over OTUs for years (https://doi.org/10.1038/ismej.2017.119). The use of replicon sequence variants (RSV), also commonly called amplicon sequence variants (ASV), is superior to OTUs due to the ability for finer resolution. Using the SILVA database at 99% sequence homology is superior to having to impose an arbitrary dissimilarity threshold that defines molecular OTUs, typically at a 97% threshold. As stated, using a 99% threshold does allow finer resolution, but that resolution is limited by the targeted variable region’s gene sequence and whether there are enough differences in the sequence between taxa in that variable region to differentiate taxa at the species level. This will hold true for some taxa and not for others.

Reviewer 2 Report

The manuscript by Cernioglo et al. describes the outcomes of a randomized, double-blind, placebo-controlled trial assessing the effect of Streptococcus salivarius K12 (S. salivarius) supplementation on oral S. salivarius abundance in health adults. To date, no double-blind, placebo-controlled trials have been conducted making this a valuable addition to the literature.  The manuscript is well-written but I have some points below that require clarification to improve the manuscript.

Comments:

  1.  The major concern I have is the potential for readers to miss the point that it was a probiotic cocktail that was given and not just S. salivarius. The probiotic powder also contained L. acidophilus and B. lactis, in addition to a prebiotic, inulin. I believe it is important that the title indicate that it was “Probiotic supplementation with a cocktail containing S. salivarius…”
  2. It is also important to state in the Methods why a cocktail was given and not just S. salivarius. Please provide a justification for the other two probiotic strains that were included.
  3. The addition of 450 mg of inulin would allow the powder to be classified as a synergistic synbiotic according to the latest consensus statement on the definition of synbiotic (Nat Rev Gastroenterol Hepatol 2020;17(11):687-701) if the S. salivarius utilizes inulin. It would be good to comment on this.
  4. It would be good in line 35 to cite the original paper with the probiotic definition: “Expert consensus document. The International Scientific Association for Probiotics and Prebiotics consensus statement on the scope and appropriate use of the term probiotic. Nat Rev Gastroenterol Hepatol 2014;11(8):506-514.”
  5. There are some minor formatting issues with the numbers/heading for 2.7 and 2.8 in lines 201 and 210.
  6. In the Discussion, please indicate as appropriate that it was a mixture of probiotics that contained S. salivarius. The way the study was designed it is not possible to isolate the effects of the S. salivarius  from the other two probiotic strains and the inulin. Any conclusions should be written in a way to clarify it was not a single strain probiotic that was administered.
  7. Some of the references only have author initials and not surnames.

Author Response

Reviewer 2

The manuscript by Cernioglo et al. describes the outcomes of a randomized, double-blind, placebo-controlled trial assessing the effect of Streptococcus salivarius K12 (S. salivarius) supplementation on oral S. salivarius abundance in health adults. To date, no double-blind, placebo-controlled trials have been conducted making this a valuable addition to the literature.  The manuscript is well-written but I have some points below that require clarification to improve the manuscript.

Comments:

  1.  The major concern I have is the potential for readers to miss the point that it was a probiotic cocktail that was given and not just S. salivarius. The probiotic powder also contained L. acidophilus and B. lactis, in addition to a prebiotic, inulin. I believe it is important that the title indicate that it was “Probiotic supplementation with a cocktail containing S. salivarius…”

The reviewer’s comments mirror and expand on Reviewer 1’s point of view, and we have addressed some aspects already in response to those Reviewer 1’s comments. We have additionally altered the title and highlighted the multiple-strain nature of the test product in several places within the manuscript so it will not be missed.

  1. It is also important to state in the Methods why a cocktail was given and not just S. salivarius. Please provide a justification for the other two probiotic strains that were included.
    The probiotic formula supplied to us by the study sponsor contained the other two strains for commercial reasons. They were included to provide gastrointestinal benefits, not induce oral microbiome changes. We did not track the GI microbiome by analyzing feces as that was not a focus of the study, however we did track GI symptoms. We have added language to the manuscript to clarify this rationale 

  1. The addition of 450 mg of inulin would allow the powder to be classified as a synergistic synbiotic according to the latest consensus statement on the definition of synbiotic (Nat Rev Gastroenterol Hepatol 2020;17(11):687-701) if the S. salivarius utilizes inulin. It would be good to comment on this.

As we have neither investigated the ability of the three strains to metabolize inulin, shown a synergistic interaction/effect between any of the strains and the prebiotic, or shown any oral health/microbiome effects mediated by the combination (as opposed to gastrointestinal ones), we believe it would be premature to describe the product as a synergistic synbiotic for oral health. As our focus is on the oral environment, it may be premature to even describe inulin as a prebiotic in that context, as to our knowledge it has not been shown to confer an oral health benefit or impact the oral microbiome. We have added clarifying language in the discussion to make these facts clear to the reader. 

  1. It would be good in line 35 to cite the original paper with the probiotic definition: “Expert consensus document. The International Scientific Association for Probiotics and Prebiotics consensus statement on the scope and appropriate use of the term probiotic. Nat Rev Gastroenterol Hepatol 2014;11(8):506-514.”

We thank the reviewer for the suggestion and included the published work in our manuscript.

  1. There are some minor formatting issues with the numbers/heading for 2.7 and 2.8 in lines 201 and 210.

Thank you. We fixed the formatting.

  1. In the Discussion, please indicate as appropriate that it was a mixture of probiotics that contained S. salivarius. The way the study was designed it is not possible to isolate the effects of the S. salivarius  from the other two probiotic strains and the inulin. Any conclusions should be written in a way to clarify it was not a single strain probiotic that was administered.

Thank you, we believe our new version has clarified this point for the reader in several places.

  1. Some of the references only have author initials and not surnames.

Thank you. We fixed the formatting of the bibliography.